# Characterization and Genome Study of Novel Lytic Bacteriophages against Prevailing Saprophytic Bacterial Microflora of Minimally Processed Plant-Based Food Products

**DOI:** 10.3390/ijms222212460

**Published:** 2021-11-18

**Authors:** Michał Wójcicki, Paulina Średnicka, Stanisław Błażejak, Iwona Gientka, Monika Kowalczyk, Paulina Emanowicz, Olga Świder, Barbara Sokołowska, Edyta Juszczuk-Kubiak

**Affiliations:** 1Laboratory of Biotechnology and Molecular Engineering, Department of Microbiology, Prof. Wacław Dabrowski Institute of Agricultural and Food Biotechnology—State Research Institute, Rakowiecka 36 Street, 02-532 Warsaw, Poland; michal.wojcicki@ibprs.pl (M.W.); paulina.srednicka@ibprs.pl (P.Ś.); monika.akimowicz@ibprs.pl (M.K.); paulina.emanowicz@ibprs.pl (P.E.); 2Department of Biotechnology and Food Microbiology, Institute of Food Sciences, Warsaw University of Life Sciences (WULS-SGGW), Nowoursynowska 166 Street, 02-776 Warsaw, Poland; stanislaw_blazejak@sggw.edu.pl (S.B.); iwona_gientka@sggw.edu.pl (I.G.); 3Department of Food Safety and Chemical Analysis, Prof. Wacław Dąbrowski Institute of Agricultural and Food Biotechnology—State Research Institute, Rakowiecka 36 Street, 02-532 Warsaw, Poland; olga.swider@ibprs.pl; 4Department of Microbiology, Prof. Wacław Dabrowski Institute of Agricultural and Food Biotechnology—State Research Institute, Rakowiecka 36 Street, 02-532 Warsaw, Poland; barbara.sokolowska@ibprs.pl

**Keywords:** bacteriophages (phages), lytic activity, phage genome, saprophytic bacterial strains, unconventional food preservation

## Abstract

The food industry is still searching for novel solutions to effectively ensure the microbiological safety of food, especially fresh and minimally processed food products. Nowadays, the use of bacteriophages as potential biological control agents in microbiological food safety and preservation is a promising strategy. The aim of the study was the isolation and comprehensive characterization of novel bacteriophages with lytic activity against saprophytic bacterial microflora of minimally processed plant-based food products, such as mixed leaf salads. From 43 phages isolated from municipal sewage, four phages, namely *Enterobacter* phage KKP 3263, *Citrobacter* phage KKP 3664, *Enterobacter* phage KKP 3262, and *Serratia* phage KKP 3264 have lytic activity against *Enterobacter ludwigii* KKP 3083, *Citrobacter freundii* KKP 3655, *Enterobacter cloacae* KKP 3082, and *Serratia fonticola* KKP 3084 bacterial strains, respectively. Transmission electron microscopy (TEM) and whole-genome sequencing (WGS) identified *Enterobacter* phage KKP 3263 as an *Autographiviridae*, and *Citrobacter* phage KKP 3664, *Enterobacter* phage KKP 3262, and *Serratia* phage KKP 3264 as members of the *Myoviridae* family. Genome sequencing revealed that these phages have linear double-stranded DNA (dsDNA) with sizes of 39,418 bp (KKP 3263), 61,608 bp (KKP 3664), 84,075 bp (KKP 3262), and 148,182 bp (KKP 3264). No antibiotic resistance genes, virulence factors, integrase, recombinase, or repressors, which are the main markers of lysogenic viruses, were annotated in phage genomes. *Serratia* phage KKP 3264 showed the greatest growth inhibition of *Serratia fonticola* KKP 3084 strain. The use of MOI 1.0 caused an almost 5-fold decrease in the value of the specific growth rate coefficient. The phages retained their lytic activity in a wide range of temperatures (from −20 °C to 50 °C) and active acidity values (pH from 4 to 11). All phages retained at least 70% of lytic activity at 60 °C. At 80 °C, no lytic activity against tested bacterial strains was observed. *Serratia* phage KKP 3264 was the most resistant to chemical factors, by maintaining high lytic activity across a broader range of pH from 3 to 11. The results indicated that these phages could be a potential biological control agent against saprophytic bacterial microflora of minimally processed plant-based food products.

## 1. Introduction

Interest in minimally processed food is still growing, which is resulting in an extending assortment of food products of this type and also in their increased availability to consumers. The plant-based food products available on the market include germinated seeds, mixed leaf salads, ready-to-eat salads, fruits salads, juices, and veggie cocktail mixes. The production of minimally processed food involves only basic treatments that enable manufacturing ready-to-eat (RTE) food products, without compromising their natural properties [1]. Owing to the use of mild methods of heat treatment and preservation, often combined with physicochemical and biological methods, minimally processed food products should retain the sensory traits of freshness, including turgor, aroma, taste, and color [2,3,4,5]. Their production process is also expected to preserve the thermolabile nutrients (vitamins and provitamins, minerals, or phytocompounds). The packaging process of food usually makes use of conditions of modified atmosphere and special (intelligent) packages tailored to individual food products [6,7].

Considering the high microbiological contamination of raw materials, effective inhibition of bacteria growth in fresh and minimally processed products is necessary. Many research centers have undertaken attempts at the synergistic preservation of food products of plant origin to control microbial food spoilage and to extend product shelf life [8,9,10,11,12]. For instance, studies have been performed to determine the effect of hot water treatment or spraying with solutions of weak organic acids on the quality of minimally processed foods [13,14,15,16,17,18,19]. The first solution enables improving the stability of plant-based food products, while the second one can reduce counts of saprophytic microflora in products of this type. Nonetheless, the physical methods of preservation used in the food industry can fail to ensure the expected sensory properties and a sufficient level of accompanying microflora reduction in the manufactured products [20,21]. Preservation treatments lead to the degradation of the cell wall (e.g., unprocessed products: lettuces, sprouts, fresh vegetables, and fruits), which causes water activity to increase in the production environment as well as the bioactive food components [21]. In turn, the high water activity in a food product facilitates bacteria development, while the cell juice represents an excellent source of nutrients to the native flora of the product and facilitates the contact between released enzymes and substrates [2].

In recent years, great attention has been paid to biological methods of food preservation [22,23]. Recently, there has been growing interest in the use of lytic bacteriophages as an innovative bio-preservative of fresh and minimally processed food products that can inhibit the growth of food-borne pathogens and spoilage microorganisms [22,24,25]. The use of bacteriophages in the biocontrol of a variety of bacterial pathogens has been studied in raw chicken meat [26,27,28], fresh-cut fruits [29,30], and fresh products including sprout seeds [22,31,32] and lettuce [33,34]. The important feature of bacteriophages is that they do not pose a health risk to consumers because they are strain-specific bacterial viruses and do not cause damage to the gut microbiome of the host [24,35,36,37]. As biocontrol agents in the food chain, phages should have a broad host range [38], and activity in a wide range of temperatures [22] and pH values [23]; should withstand environmental stress induced by food processing conditions [39]; and should not have any pathogenic or allergic-associated properties [23,40].

Nowadays, multiple companies offer commercial bacteriophage preparations intended for the food industry. Many of these preparations have been approved by the FDA, such as PhageGuard (e.g., Listex^TM^ P100, Secure Shield E1, EcoShield^TM^, ListShield^TM^, ShigaShield^TM^, SalmoFresh^TM^), and the USDA (PhageGuard, Finalyse^®^) [41,42]. Ten commercial phage preparations have received a temporary GRAS status [43]. In 2016, the EFSA issued a report concerning the evaluation of the safety of use and efficacy in eradicating pathogenic bacteria of the Listex^TM^ P100 preparation by Micreos Food Safety [44], whose effectiveness has been confirmed by scientific research [5,36]. However, not all studies have proved the efficacy of commercial bacteriophage preparations. For example, a bacteriophage preparation produced by Finalyse^®^ did not significantly reduce the number of *Escherichia coli* O157:H7 compared to the control samples [45]. In turn, Oladunjoye et al. [46] showed an increased efficacy of the coupled use of a bacteriophage preparation with sucrose monolaurate as a compound exhibiting antibacterial activity. It should be emphasized that commercial bacteriophage preparations are mainly targeted at pathogenic bacteria that prevail in the food environment [23,41,42,43].

Food is an organic perishable substance, which is susceptible to spoilage due to microbial, chemical, or physical activities. Therefore, it is essential to ensure global food safety and preserve the quality of these food products [5]. To ensure food safety and the long shelf-life of foods, it is important to understand food spoilage mechanisms and food preservation techniques [28]. However, increasing the shelf-life of foodstuff without compromising the original food properties is still critical and challenging [40,47,48,49,50]. Fresh vegetables are among the more challenging food products to commercially produce and distribute [48,49]. Vegetables are a source of nutrients for spoilage microflora because of their near-neutral pH and high water activity [51,52], and their microbial contamination sources include raw materials and contact with processing equipment [48]. In minimally processed foods, in addition to pathogenic bacteria, the rate of growth and development of saprophytic microflora determines the shelf-life [53]. Saprophytic microflora cause food spoilage and deteriorate food quality, and thereby shorten its shelf-life [22,23]. Therefore, the use of lytic phages to eliminate spoilage bacteria during the storage of minimally processed foods may bring the expected results. Contrary to conventional methods of food preservation, the use of such food protection does not change the physical properties of the food (e.g., texture, color, elasticity) [22,23,48]. Furthermore, recent technological advances have seen next-generation sequencing (NGS) become increasingly used in phage research, providing a more detailed genome phage characterization, screening of unfavorable genes, and evaluation of potentially useful gene products [54,55].

Thus, the aim of this study was the isolation and characterization of lytic bacteriophages against prevailing saprophytic bacteria of minimally processed fresh plant-based food products. In our study, bacteriophages were isolated from pre-treated municipal sewage, which offers optimal conditions for phages development due to its strong contamination with bacteria. The whole-genome sequencing (WGS) approach was utilized for the characterization of the phage genome sequence analysis and annotation. The stability of isolated phages was tested at acidic and alkaline pH, at high temperatures, and in cold storage. Finally, their lytic activity against prevailing saprophytic bacteria strains isolated from leaf salads and mixed leaf salads, mixed leaf salad with carrot, and mixed leaf salad with beetroot from retail sales was evaluated.

## 2. Results and Discussion

### 2.1. Microbiological Analysis of the RUC, MLSC, and MLSB Products

Pure bacterial cultures were isolated during the evaluation of the microbiological quality of the tested products: RUC, MLSC, and MLSB, respectively. The selected morphological and physiological features of four bacterial strains are presented in Table 1. Bacteriophages with strong lytic properties against prevailing bacterial strains in tested RUC, MLSC, and MLSB were isolated and characterized in the subsequent stages of the study.

### 2.2. Evaluation of the Lytic Activity of Phages against Bacterial Hosts

The titer of selected bacteriophages after their proliferation in host bacterial cells was determined using the conventional inoculation with bacteria on double-layered agar (Figure 1).

Bacterial cell lysis caused by *Enterobacter* phage KKP 3262, *Enterobacter* phage KKP 3263, and *Serratia* phage KKP 3264 (Figure 1B–D) resulted in the formation of plaques with a characteristic “halo” zone, which indicates the lysis of the bacterial host cells. Infection of the capsular bacterial cell by bacteriophages depends on the breakdown of the capsular polysaccharide layer [56,57,58]. Phage enzymes such as endolysins and holins are associated with the degradation of the cell envelope during bacteriophage infection release of daughter virions [59,60,61,62,63]. These enzymes diffuse into the substrate, deprive the cell sheaths of the cells growing around the plaques, and appear as bright zones called “halo” zones [63,64]. Research shows that these enzymes reduce the mechanical strength and resistance of the bacterial cell wall, which causes bacteriolysis and the release of daughter virions [65,66]. Due to the limited activity and functionality of individual endolysins, the group of these enzymes does not belong to the same type, and the process of lysis of the bacterial cell caused by phage infection may be different for different bacteria [67]. Among the tested phages, plaques of *Enterobacter* phage KKP 3263 against *Enterobacter ludwigii* KKP 3083 had the largest diameter. This is related to the size of the phage [68]. After incubation, plaques formed were counted, and their number was converted, including the dilution factor, into phage titer in the lysate. The concentrations of studied phages against the isolated bacterial strains are presented in Table 2.

The lytic activity of phages was determined using a Bioscreen C automatic growth analyzer. To this end, growth curves were plotted for each strain (optical density in the function of bacteria cell number). Determination of an equation of a straight line allowed the determination of the optical density values for bacterial strain cultures (Table 2), which in turn enabled the adjustment of the appropriate values of the multiplicity of infection coefficient for each phage (MOI 1.0 and 0.1, respectively). Optical density measurement made with the Bioscreen C automatic growth analyzer allowed us to establish the onset and duration of the logarithmic stage of growth of the bacterial strains which were deliberately infected with specific phages at MOI 1.0 and 0.1, respectively, compared to the control culture. The changes in the optical density of the tested bacterial strains after incubation with specific phages are presented in Table 3. The lower coefficients of the specific growth rate determined for the phage-infected samples point to significant suppression of cell division during the logarithmic stage of growth of the tested strains. Results showed that infection of the tested bacterial strains at MOI 1.0 more strongly inhibited cell divisions compared to the lower infection coefficient (MOI 0.1) (Figure 2).

Infection of *Citrobacter freundii* KKP 3655 by *Citrobacter* phage KKP 3664 at MOI 1.0 caused an almost 2-fold decrease in the value of the specific growth rate coefficient, whereas infection of the *Enterobacter ludwigii* KKP 3083 by *Enterobacter* phage KKP 3263 caused over a 2-fold decrease of this coefficient, compared to the control sample. *Serratia* phage KKP 3264 specific to the *Serratia fonticola* KKP 3084 showed the greatest growth inhibition of bacterial hosts. The use of MOI 1.0 caused an almost 5-fold decrease in the value of the specific growth rate coefficient, whereas MOI 0.1 caused an over 2-fold decrease of this coefficient compared to the control sample. In general, the infection with phage at MOI 1.0 had a stronger inhibiting effect on the growth of bacterial cultures than the infection at MOI 0.1.

The rapid growth rate of culture in the logarithmic phase was observed for the control of the *Citrobacter freundii* KKP 3655, *Enterobacter cloacae* KKP 3082, and *Enterobacter ludwigii* KKP 3083 after 4 h 15 min, 3 h 30 min, and 1 h 45 min, respectively (Figure 2A–C). Duration of this growth phase was different between individual strains and reached 10 h for *Citrobacter freundii* KKP 3655, 14 h 30 min for *Enterobacter cloacae* KKP 3082, and 17 h 30 min for *Enterobacter ludwigii* KKP 3083, respectively. For *Citrobacter freundii* KKP 3655 and *Enterobacter cloacae* KKP 3082, infection with the *Citrobacter* phage KKP 3664 and *Enterobacter* phage KKP 3262, respectively, at MOI 1.0 was the most effective in delaying the onset of the log-phase, which began after 12 h and 7 h 24 min, respectively. At MOI 1.0, the logarithmic growth phase of the *Citrobacter freundii* KKP 3655 lasted 11 h 30 min, and the OD_600_ value of the culture increased by 0.136. In the case of the *Enterobacter cloacae* KKP 3082, the log-phase was longer (14 h 45 min) and increased OD_600_ by 0.147 (Table 3).

Infection of the *Enterobacter ludwigii* KKP 3083 by *Enterobacter* phage KKP 3263 at MOI 0.1 delayed the onset of the log-phase compared to MOI 1.0. At MOI 0.1, its log-phase lasted 13 h 15 min, and the optical density of its culture increased by 0.206, whereas at MOI 1.0 the lag-phase began after 5 h 45 min and lasted 16 h 15 min, causing culture optical density increase by 0.160 (Table 3). In the case of the control *Serratia fonticola* KKP 3084, the log-phase did not begin earlier compared to the phage-infected culture (Figure 2D). Strain infection by *Serratia* phage KKP 3264 at MOI 1.0 accelerated the onset of the log-phase (after 1 h 15 min vs. 5 h 30 min in the control), which lasted 21 h 15 min and caused optical density increase by 0.102. In the *Serratia fonticola* KKP 3084 infected at MOI 0.1, the log-phase began after 2 h 15 min and lasted 13 h 30 min. Within this period, the optical density of the culture increased by 0.173 (Table 3).

Changes in the onset of the logarithmic growth phase in phage-infected cultures were reported by Zhao et al. [69]. The authors showed that, regardless of the multiplicity of infection, the bacterial cultures treated with phages began the log-phase significantly later compared to the control. In the study conducted by Mahmoud et al. [70], the growth of *Salmonella* Kentucky infected with bacteriophages at MOI 1.0 was delayed by all tested phages compared to control cultures. Moreover, complete inhibition of the bacterial growth was observed after 24 h incubation with tested phages. In the experiment carried out by Yu et al. [71], the phage-infected culture of *Pseudomonas syringae* pv. *actinidiae* exhibited poorer growth compared to the control culture up to 24 h. In the subsequent 24 h, part of the bacterial culture showed a stronger growth compared to the control, indicating that bacterial cells very likely gained tolerance or resistance to this bacteriophage.

### 2.3. TEM Morphology of Selected Bacteriophages

The morphology of the bacteriophages was determined using transmission electron microscopy (TEM) in differential staining (Figure 3). Due to the presence of both the head and the tail, phages were assigned as representatives of the order *Caudovirales*, that is, bacteriophages with a complex structure [72]. Based on TEM analysis, phages were divided into two families, *Myoviridae* and *Autographiviridae*. The *Citrobacter* phage KKP 3664, *Enterobacter* phage KKP 3262, and *Serratia* phage KKP 3264 against *Citrobacter freundii* KKP 3655, *Enterobacter cloacae* KKP 3082, and *Serratia fonticola* KKP 3084 have been classified as the *Myoviridae* family (Figure 3A,B,D). Bacteriophages of this family had a long, stiff, and contractile tail [70,72,73]. The tail of the *Citrobacter* phage KKP 3664 was 107.7 nm long, that of the *Enterobacter* phage KKP 3262 was 115.4 nm long, and the *Serratia* phage KKP 3264 was 123.8 nm long. The tails were composed of the inner tube and a noticeable outer, helical, contractile sheath. The baseplate and the tail tube were visible at the end of the tail (Figure 4). The tails were connected to the capsid with a neck. The phages heads, 69.2 nm, 92.3 nm, and 76.2 nm in diameter, respectively, had an icosahedral structure. An additionally stained (with iron (II) chloride) nucleic acid present in the capsid is shown in Figure 3D.

The *Enterobacter* phage KKP 3263 against *Enterobacter ludwigii* KKP 3083 belonged to the *Autographiviridae* family (podophages) [74] (Figure 3C). TEM analysis showed common *Autographiviridae* morphological characteristics, resembling the *Podoviridae* family [75,76,77], with an isometric icosahedral head (head diameter: 57.7 nm) and a short non-contractile tail stub without appendages or fibers (tail length: 46.2 nm). The TEM analysis showed that among the tested bacteriophages, *Enterobacter* phage KKP 3263 phages had the smallest size (103.9 nm); therefore, they diffused most easily in soft agar and more effectively infected neighboring bacteria (see the largest plaques in Figure 1). Once again, ICTV has recently updated the taxonomy system with changes to the order of phages, families, subfamilies, genus, and species [75,77]. Most of the bacteriophages characterized so far (96%) have been classified to the order *Caudovirales,* and most of them are representatives of the *Myoviridae* or *Siphoviridae* families [78]. According to the literature, most of the bacteriophages from the *Siphoviridae* family are moderate (lysogenic) phages [79]. Moreover, the length of the tail indicates phage stability in the environment [73,80]. Bacteriophages with short tails (e.g., podophages) or without tails are, usually, more resistant in the environment, whereas those with long tails (e.g., myophages or siphophages) are more prone to damages, which leads to the loss of their antibacterial activity [73,81]. The tailed phages contain a genome in the form of two-stranded DNA [73].

The TEM images indicated that there is no relationship between the phage taxonomy and the staining method. Phages from the *Myoviridae* family were stained with both 2% phosphotungstic acid and 2% uranyl acetate (Figure 3A,B). Moreover, the nucleic acid of these phages can be contrasted with 0.5% iron (II) chloride (Figure 3D). In turn, the staining of the phages from the *Autographiviridae* family was more effective with a 2% phosphotungstic acid solution (Figure 3C).

### 2.4. Analysis of Phage Genomes

The complete genomes of the *Enterobacter* phage KKP 3263, *Citrobacter* phage KKP 3664, *Enterobacter* phage KKP 3262, and *Serratia* phage KKP 3264 have been sequenced and deposited in the GenBank database under the accession numbers OK210074, OK210075, OK210076, and OK210077, respectively. Moreover, the phages were deposited in the Polish Collection of Industrial Microbial Cultures as the partner of Microbial Resource Research Infrastructure (MIRRI).

Similar to TEM, genome analysis confirmed that one phage belonged to the *Autographiviridae,* and three were classified for the *Myoviridae* family. Proteomic trees generated by the BIONJ-program-based [82] TBLASTX genomic sequence comparisons of other phage genomes deposited in the Virus-Host DB [83] are presented in Figure 5.

Genome sequencing of the *Enterobacter* phage KKP 3263, *Citrobacter* phage KKP 3664, *Enterobacter* phage KKP 3262, and *Serratia* phage KKP 3264 revealed that these phages have linear double-stranded DNA (dsDNA).

A genome map of *Enterobacter* phage KKP 3263 belonging to the *Autographiviridae* is presented in Figure 6. *Enterobacter* phage KKP 3263 has a genome length of 39,418 bp with a total G + C content of 52%. According to the sequence analysis, the genome of the *Enterobacter* phage KKP 3263 contains 50 predicted open reading frames (ORFs), all of them situated on the positive strand (Figure 6 and Appendix A); no potential tRNA coding genes were found in the *Enterobacter* phage KKP 3263 genome. Among the 50 ORFs, twenty predicted proteins have known potential functions responsible for DNA replication/transcription/packaging, cell lysis, and phage morphology, while the remaining 30 ORFs were annotated as hypothetical proteins.

BLASTn alignment of the *Enterobacter* phage KKP 3263 nucleotide sequences with the previously sequenced phages (Appendix A) showed an 87% degree of identity with the *Escherichia* phage vB_EcoP_SP7 (GenBank Acc. No. MT682707.1). The structures of *Autographiviridae* phages belonging to different taxa have much in common [86]. Meanwhile, the genome analysis of *Autographiviridae* phages showed that the majority of protein sequences differ substantially, except for the terminase large subunit (TerL) and major capsid protein, which are more conservative [86]. Comparison of the TerL amino acid sequences of *Enterobacter* phage KKP 3263 with *Autographiviridae* phages is presented in Figure 7. Analysis showed that *Enterobacter* phage KKP 3263 had high TerL similarity with *Citrobacter* phage CR44b (96.1%; NC_023576) and also with *Escherichia* phage Ro45lW (94.9%; NC_048136) and *Citrobacter* phage SH3 (94.6%; NC_031123) (Figure 7).

Sequencing of the *Serratia* phage KKP 3264, *Enterobacter* phage KKP 3262, and *Citrobacter* phage KKP 3664 genomes confirmed that these phages are members of the *Myoviridae* family. BLASTn alignment between these phages showed 8% sequence similarity only between *Enterobacter* phage KKP 3262 and *Citrobacter* phage KKP 3664 (not shown). The results showed that *Serratia* phage KKP 3264 has the longest genome, containing 148,182 bp with a C+G content of 41%. Its genome encodes 236 ORFs on both strands and 16 tRNA localized in the genomic region between 61–63 and 65–77 ORFs, respectively (Figure 8 and Appendix A).

Putative functions were assigned to 7 ORFs classified as structural proteins (major capsid protein, tail fiber assembly protein U), DNA synthesis/replication/repair proteins (ribonucleoside-diphosphate reductase large subunit, thymidine kinase, anaerobic ribonucleoside-triphosphate reductase), and lysis proteins (endolysin); however, 229 ORFs encoded in *Serratia* phage KKP 3264 genome were annotated as hypothetical proteins. BLASTn similarity searches performed for *Serratia* phage KKP 3264 and related phages deposited in GenBank revealed a 99.03% nucleotide similarity with *Escherichia* phage vB_EcoM_PHB05 (NC_052652) and 98.75% with *Escherichia* phage vB_vPM_PD06 (NC_052653) (Appendix A). Similarly, TBLASTX genomic sequence comparisons of other phage genomes deposited in the Virus-Host DB [83] showed high similarity of *Serratia* phage KKP 3264 with four of the *Myoviridae* phages: *Escherichia* phage vB_vPM_PD06 (97.7%; NC_052653), *Escherichia* phage vBEcoM_PHB05 (97.6%; NC_052652), and *Escherichia* phage alia (97.5%; NC_052655) (Figure 9).

According to the sequence analysis, the *Enterobacter* phage KKP 3262 genome consists of 84,075 bp dsDNA with 165 ORFs and total C+G content of 39.5%. Among the 165 ORFs, 36 had significant homology to reported functional genes, while the remaining 129 ORFs were annotated as hypothetical proteins (Figure 10 and Appendix A).

In addition, 17 ORFs were in the plus strand and 148 were in the minus strand. The proteins encoded by *Enterobacter* phage KKP 3262 could be divided into several functional modules: phage structure/assembly (ORFs 7, 146, 160, 161, 164 and 149, 150, 153–159), DNA replication/modification/regulation (ORFs 8, 12, 15, 18, 19, 21, 22, 30, 41), phage lysis (ORFs 95), phage packing (ORF165), and some additional proteins. Moreover, the presence of the 19 tRNA genes between 112–115, 121, and 123–137 ORFs, respectively, were identified (Figure 10 and Appendix A). The presence of tRNA genes in phage genomes, especially in virulent phages, is a common phenomenon [87,88]. The discovery of tRNA has been reported for the first time in the genome of the T4 myovirus infecting *Escherichia coli,* and extensive studies for expression and functionality of T4 tRNA [89] and other phages [87,90,91] have been reported. Some studies have proposed that tRNA-containing phages have a codon bias that diverges from that of the bacterial host, therefore using the tRNAs to compensate for a metabolic difference [91,92]. Canchaya et al. [93] suggested that the presence of tRNA in the phage genome can be correlated with better integration of virulent phages inside the host chromosome. Bailly-Bechet et al. [94] reported that there is a positive association between the size of the phage genome and the number of tRNA genes it contains. Similar results were also reported by Morgado and Vincente [90], stressing again the correlation between the number of tRNA genes and genome length. The presence of tRNAs was predicted in ssDNA (single-stranded DNA) and ssRNA (single-stranded RNA) viruses as well in dsDNA phage genomes of families from *Caudovirales* order [94]. Lack of tRNA genes can be associated with highly compact genomes of phages that tend to lack any translational-associated genes to exclude nonessential information [94]. BLASTn analysis revealed that *Enterobacter* phage KKP 3262 genome had high similarity with two of the *Myoviridae* family phages (Appendix A): *Klebsiella* phage vB_KaeM_KaAlpha (96%; GenBank Acc. No. MN013084.1) and *Enterobacter* phage PG7 (95%; GenBank Acc. No. KJ101592.1). Moreover, based on the results of TBLASTx analysis, the protein sequences of the *Enterobacter* phage KKP 3262 displays similarity to three phages: *Cronobacter* phage Pet-CM3-4 (NC-055725; 95.2%), *Enterobacter* phage PG7 (NC_023561; 94.9%), and *Enterobacter* phage CC31 (NC_0146662; 93.5%) (Figure 11).

Analysis of the *Citrobacter* phage KKP 3664 genome (61,608 bp) showed 79 ORFs and no tRNAs (Figure 12 and Appendix A). The overall G+C content of its genome is 43%.

BLASTn search (Appendix A) identified a 96% nucleotide similarity of the *Citrobacter* phage KKP 3664 genome with previously sequenced *Citrobacter* phage vB_CfrM_CfP1 (GenBank Acc. No. KX245890.1), *Citrobacter* phage Miller (GenBank Acc. No. KM236237.1), and *Buttiauxella* phage vB_ButM_GuL6 (GenBank Acc. No. MT334653.1). Moreover, as shown in Figure 13, the TBLASTX analysis showed that protein sequences of the *Citrobacter* phage KKP 3664 have 91.5% similarity with other virulent *Myoviridae* phages (*Escherichia* virus RB43, NC_007023; *Escherichia* phage RB16, HM134276; *Escherichia* virus RB16; NC_014467) that infect Enterobacteriaceae. Within the total ORFs, *Citrobacter* phage KKP 3664 has 29 ORFs on the leading strand, and 50 ORFs on the complementary strand. BLASTp analysis identified 31 predicted proteins with putatively known functions, among which tail and capsid assembly proteins (capsid proteins, ORF48, 50; tail proteins, ORF34, 42-43; neck proteins, ORF36-37; baseplate wedge proteins, ORF25, 28-33), DNA replication (ATP-dependent DNA helicases, ORF53-54; DNA adenine methylase, ORF75; deoxynucleotide monophosphate kinase, ORF20), and DNA packaging (terminase proteins, ORF40-41) were annotated (Appendix A). No tRNAs were found in the phage genome, suggesting that the *Citrobacter* phage KKP 3664 did not take over the host transcription/translation system, but uses host tRNAs for the synthesis of phage proteins [95,96].

Currently, bacteriophages are frequently used for inactivation and control of food-borne pathogens—such as *Salmonella*, *Escherichia coli* O157:H7, *Listeria*, and *Campylobacter*—in various foods, ranging from ready-to-eat deli meats to fresh fruits and vegetables [97,98,99]. For biocontrol applications in the food industry, strictly lytic phages are used to avoid potential threats related to virulence factors that are associated with lysogenic phages [41,100,101]. In our study, the BLASTp analysis showed that the genomes of *Enterobacter* phage KKP 3263, *Citrobacter* phage KKP 3664, *Enterobacter* phage KKP 3262, and *Serratia* phage KKP 3264 do not contain sequences of genes encoding integrase, recombinase, repressors, or excisionase, which are the main markers of lysogenic viruses [102]. Our results indicated that these phages should be considered as strictly lytic (virulent) phages. Overall, the genetic analysis suggests that these phages would be a safe biocontrol agent for food application [38]. On the other hand, it should be remembered that phage transduction is one of the mechanisms of horizontal gene transfer (HGT) between bacterial cells [103,104]. Therefore, bacteriophages can be potential sources of virulence factors and antibiotic resistance of bacteria [105]. Furthermore, antibiotic resistance genes (ARGs) carried by phages are considered especially threatening due to their prolonged persistence in the environment, fast replication rates, and ability to infect diverse bacterial hosts [106]. In this study, no ARGs were annotated in the phages’ genomes. Moreover, genes associated with the virulence factors were also absent.

It has been reported that phages of the *Myoviridae* family are associated with the lytic life cycle and characterized by the high efficiency of the bacterial lysis [107]. In *Serratia* phage KKP 3264 and *Enterobacter* phage KKP 3262 belonging to *Myoviridae*, the host lysis system consists of the predicted ORF11 (Figure 8 and Appendix A) and ORF95 (Figure 10 and Appendix A), respectively, encoding putative endolysin. In the *Enterobacter* phage KKP 3263 genome, belonging to *Autographiviridae,* the presence of the ORF35 encoding endolysin was also annotated (Figure 6 and Appendix A). The function of endolysin is to lyse host cells and release phage progeny [108]. At the final stage of the life cycle, the dsDNA phages use the endolysin and holin to lyse the host bacterium by degrading the inner membrane and peptidoglycan layer of the bacterial membrane, respectively [109]. Phage-encoded proteins such as endolysins, exopolysaccharidases, and holins proved their ability as promising alternative antibacterial products [107,108].

### 2.5. Effect of Selected Environmental Factors on the Preservation of the Lytic Activity of Phages

In this study, the lytic activity of isolated phages on exposure to a wide range of temperatures (from −20 to 80 °C) and the active acidity (pH from 3 to 12) values of the medium was evaluated.

#### 2.5.1. Influence of Temperature on the Lytic Activity of Phages

Temperature is the crucial factor for bacteriophages’ viability and activity in the environment [110,111], playing a crucial role in the attachment, penetration, and amplification of phage particles in their host cells [112]. The effect of different temperatures on the lytic activity of studied phages is presented in Figure 14. The lytic activity was expressed as phage titer (in PFU mL^−1^) and assumed to reach 100% for the control parameters.

Regarding the *Citrobacter freundii* KKP 3655 (Figure 14A), the highest lytic activity was determined for the *Citrobacter* phage KKP 3664 exposed to temperatures from −20 to 50 °C. A significant decrease in the phages’ activity and reduction of phages’ titer by 90% was observed at 60 °C compared to the control culture at 20 °C (*p* < 0.05). Pasteurization at 70 °C and above completely inactivated the lytic activity of the phage, which could be due to virion proteins and specific lytic enzymes’ degradation [113,114].

The highest lytic activity against the *Enterobacter cloacae* KKP 3082 (Figure 14B) was noticed for the *Enterobacter* phage KKP 3262 exposed to the temperature of −20 °C (*p* < 0.05). High lytic activity of this phage was also exhibited at temperatures ranging from 4 to 30 °C. In turn, *Enterobacter* phage KKP 3262 held at 40 °C showed a significantly lower (*p* < 0.05) lytic activity compared to that held at 50 °C. In the case of certain lysogenic phages (capable of the latent developmental cycle), their heat treatment (42 °C) causes the induction of prophages, which may, in turn, contribute to stronger lysis at a higher temperature of culture incubation [115,116]. The increase of temperature from 50 to 60 °C caused a reduction of phage titer by almost 3 log units (99.9%) compared to the control parameters (*p* < 0.05).

The highest lytic activity against the *Enterobacter ludwigii* KKP 3083 (Figure 14C) was noticed for the *Enterobacter* phage KKP 3263 exposed to a temperature of −20 °C (similar to *Enterobacter* phage KKP 3262 against *Enterobacter* cloacae KKP 3082). High lytic activity of *Enterobacter* phage KKP 3263 was also noticed at temperatures ranging from 4 to 50 °C. On the other hand, the increase of temperature from 50 to 60 °C and 70 °C caused a reduction of phage titer by 3 log units (99.9%) and 5 log units (99.999%), respectively, compared to the control parameters. No lytic activity of the tested phage was observed at 80 °C.

*Serratia* phage KKP 3264 against the *Serratia fonticola* KKP 3084 (Figure 14D) retained high lytic activity at 50 °C, but increasing the temperature above 50 °C resulted in a significant reduction in phage activity (*p* < 0.05), similar to the *Enterobacter* phage KKP 3263 against *Enterobacter ludwigii* KKP 3083.

In the study by Shahin and Bouzari [117], phages against *Shigella flexneri* retained their lytic activity in a temperature range from 4 to 60 °C, while at 70 °C and higher, their activity was completely inhibited. In the study conducted by Thung et al. [118], bacteriophages against *Salmonella* Enteritidis exhibited activity at a temperature of 65 °C; however, their titer decreased. It is believed that at low temperatures, only a part of the available phage’s genetic material pervades into bacterial host cells (as most bacteria multiply more slowly) and therefore fewer phage particles can be involved in the multiplication phase [119,120]. On the other hand, high temperatures can promote an extended phage latency period [119]. In our study, the low temperature did not influence inhibition of the lytic activity of tested phages. In the case of the phages against *Listeria monocytogenes*, a significant decrease in activity was observed at 37 °C [121]. It is known that host receptors for bacteriophages adsorption are located on the bacterial flagellum [122,123,124]. In *Listeria monocytogenes,* the temperature of 37 °C inhibits the expression of genes involved in the motility of bacterial flagellum (*L. monocytogenes* is motile at 25 °C, and non-motile at 37 °C) [121,125]. In the study conducted by Jamal et al. [126], the phages against *Pseudomonas aeruginosa* were stable at temperatures ranging from 37 to 65 °C, whereas the bacteriophages became completely ineffective at 70 °C. In turn, Mahmoud et al. [70] reported that the phages isolated against *Salmonella* serovars were stable in a temperature range from 30 to 70 °C, and remained active for 15 min at 80 °C.

#### 2.5.2. Influence of pH on the Lytic Activity of Phages

In this study, pH values ranging from 3 to 12 were tested. The effect of pH on the lytic activity of studied phages is presented in Figure 15A–D. Results showed that *Citrobacter* phage KKP 3664 against *Citrobacter freundii* KKP 3655 exposed to the acidity of pH 8 retained the highest activity. Incubation at extreme values of active acidity (strongly acidic or alkaline) reduced phage titer by one (pH 3 and pH 11) or two (pH 12) log units compared to the control (Figure 15A). For the *Enterobacter cloacae* KKP 3082, the highest lytic activity was noticed for the *Enterobacter* phage KKP 3262 held in a pH range from 4 to 10. Moreover, incubation at extreme values of active acidity reduced these phages’ titer by 99.0% (pH 11) or 99.9% (pH 3 and pH 12) compared to the control (Figure 15B). *Enterobacter* phage KKP 3263 against *Enterobacter ludwigii* KKP 3083 retained activity in the pH range from 4 to 11. We observed that at pH 3, its activity decreased by almost 5 log units compared to the control (Figure 15C). Our results showed that *Serratia* phage KKP 3264 against *Serratia fonticola* KKP 3084 was the most resistant to chemical factors; it retained lytic activity in the pH range from 3 to 11 (Figure 15D). Both viruses, *Enterobacter* phage KKP 3263 and *Serratia* phage KKP 3264, showed no activity at pH 12. Several studies showed that the pH stability of many phages ranged from 3 to 11, and above pH 11, phages rarely maintain active [70,127]. Shahin and Bouzari [117] reported that phages against *Shigella flexneri* retained the highest lytic activity in a medium pH range from 7 to 11, and their activity was completely inhibited by active acidity below pH 5 and above pH 13. In the experiment carried out by Jamal et al. [126], phages against multi-drug-resistant *Pseudomonas aeruginosa*-2995 were active at pH from 3 to 11, whereas at pH 1, inactivation of phages was observed. On the other hand, Li et al. [128] reported that virulent JN01 phage against *E. coli* O157:H7 in milk and beef maintained stable activity after being exposed to pH 13 for 1 h, indicating its strong alkali-resistance. Furthermore, Yu et al. [71] reported differences in the extent of tolerance of individual phages against the same bacterial strain to varied values of active acidity. Results showed that the acidic medium more strongly inhibited phages’ activity in comparison to the alkaline medium. In turn, investigations into the activity of phages against *Salmonella* Enteritidis [118] showed their complete inactivation at pH 3. Similar findings have been reported by Sváb et al. [129] and by Thung et al. [130]. In the study by Międzybrodzki et al. [131], phages retained their activity in a pH range from 5 to 9 during incubation at 37 °C. In turn, according to Wang and Sabour [132], the optimal pH for most of the bacteriophages ranges from 5 and 8. However, a drop in temperature widens their tolerance to a pH value of 4 to 10. Medium with too low pH probably affects denaturation of virion proteins, whereas survivability in a wide range of pH values is a trait desired in biocontrol [118]. In our study, bacteriophages exhibited activity in a wide range of both pH and temperature values, and therefore could be effectively used in food biocontrol.

## 3. Materials and Methods

### 3.1. Isolation and Taxonomic Identification of the Bacterial Strains from Food Products

The bacterial strains used in this study were originally isolated from commercial minimally processed leaf salads and mixed leaf salads such as rucola (RUC), mixed leaf salad with carrot (MLSC), and mixed leaf salad with beetroot (MLSB). The microbiological quality of these products was investigated in 0, 4, 7 days during cold storage at 4 °C (unpublished data). The bacterial strains were characterized based on their morphological and biochemical properties. Moreover, taxonomic identification of bacterial strains was performed using both 16S rDNA gene sequencing and MALDI-TOF analysis.

### 3.2. Bacteriophage Isolation, Purification, and Propagation

A total of 25 mL of municipal sewage was centrifuged at 10,000× *g* (20 °C for 10 min) to separate organic and mineral particles from bacteria and potential bacteriophages. The supernatant was filtered using a syringe filter with a membrane pore diameter of 0.22 μm. Then, 20 mL of the filtrated supernatant containing bacteriophages from the sewage were transferred to 20 mL of double-concentrated T-broth (composition: 8.0 g L^−1^ of enriched broth, 5.0 g L^−1^ of peptone, 5.0 g L^−1^ of sodium chloride, and 1.0 g L^−1^ of glucose). The culture medium with bacteriophages was inoculated with 1 mL of an overnight culture of a bacterial strain on a TSG medium and incubated at 37 °C for 24 h. Afterward, the culture was centrifuged at 8000× *g* for 10 min to separate bacteria from the proliferated bacteriophages. The supernatant was filtered using a syringe filter with a membrane pore diameter of 0.45 μm (according to Mirzaei and Nilsson procedure [133] with modification) and freeze-stored (−80 °C) with 20% addition of glycerol.

### 3.3. Evaluation of the Lytic Activity of Phages against Bacterial Hosts

The lytic activity of bacteriophages against the isolated bacterial strains was determined with a double-layered plate according to the method of Jamal et al. [126]. Glass tubes containing dissolved LCA were placed in a water bath at 48 °C until the temperature was equilibrated. Then, 100 μL of 0.025 M CaCl_2_ and 0.025 M MgSO_4_ were added to sterile test tubes and incubated with 100 μL of overnight bacterial culture. Then, 500 μL of diluted phage lysate were transferred to test tubes. After 20 min, the tubes were supplemented with 5 mL of soft LB agar (48–52 °C), mixed, and then poured onto the first agar layer and incubated at 30 °C. The results were read after 24 h. The unit used was plaque-forming unit per mL (PFU mL^−1^). Single-phage plaques were cut with a scalpel and purified in SM buffer according to the method proposed by Mirzaei and Nilsson [133]. Purification was performed in four rounds of single-plaque passage to ensure that the isolate represented the clonal phage population. In addition, after filtering through a syringe filter with a membrane pore diameter of 0.45 μm, each lysate was inoculated to check for possible contamination with bacterial cells. Growth curves were made for each of the bacterial strains. For this purpose, the harvested culture was inoculated on PCA medium every hour for 24 h, and the optical density was measured simultaneously. The dependence of the optical density on the number of bacterial cells was determined (performed in triplicate). Next, coefficients of the specific growth rate (μ) were computed for each strain using the following formula:μ = (ln OD_max_ − ln OD_min_)/t, (1)
where ln OD_max_—natural logarithm of the maximal value of the optical density of the culture during exponential growth phase; ln OD_min_—natural logarithm of the minimal value of the optical density of the culture during exponential growth phase; t—duration of the exponential growth phase, (h).

Once phage titer was determined and bacterial host growth curves were plotted, bacteria growth kinetics was measured using a Bioscreen C automatic growth analyzer (Yo AB Ltd., Growth Curves, Helsinki, Finland). Bacteria were proliferated in the LB broth (composition: 10.0 g L^−1^ of peptone, 10.0 g L^−1^ of sodium chloride, and 5.0 g L^−1^ of yeast extract). The culture was diluted at a ratio of 1:100 in a fresh culture medium with the addition of CaCl_2_ and MgSO_4_, both having final concentrations of 0.01 M. To ensure the optimal value of the multiplicity of infection (MOI) coefficient, flasks with the new culture were incubated at a temperature of 37 °C with continuous shaking until the desired optical density depended on the phage titer. Then, 180 μL of each culture was pipetted into multi-well plates and incubated in a Bioscreen C at 37 °C until optical density increased by OD_600_ ~0.1 compared to the control medium. Phage lysates were prepared so that the value of MOI coefficient reached 1.0 and 0.1, respectively. Then, 20 μL of respective phage lysates were added to wells, left at 20 °C for 15 min to allow the phages to adsorb to the host cell surface, and incubated at 37 °C for 24 h. The apparatus measured optical density automatically every 15 min, at a wide band of wavelengths ranging from 420 to 580 nm, with 15 s shaking preceding each readout. The test was performed in 10 replicates for each strain and infection rate.

### 3.4. Determination of Morphological Features of Selected Phages

Transmission electron microscopy (TEM) was used for determining morphological features of the isolated bacteriophages and classifying them into respective families. Propagated phage lysates were centrifuged at 4 °C and 14,500× *g* for 40 min. The excess culture medium was removed, and the pellet was suspended in 2 mL of 100 mM cold ammonium acetate (filtered through a syringe filter with a membrane pore diameter of 0.22 μm). The precipitate was disintegrated with a tip and centrifuged again. The whole procedure was repeated four times. After centrifugation, the precipitate was flushed from the Eppendorf tube wall with 50 μL of ammonium acetate according to Ackermann’s procedure [134] with modification. Then, 2 μL of the phage suspension in ammonium acetate was coated onto carbon-sputtered copper-wolfram mesh grids. After one hour, the specimen was stained for 30 sec, and excess dye was removed with a filter paper. Depending on phage type, 2% phosphotungstic acid solution neutralized to pH 7.2 with 1 M potassium hydroxide solution, 2% uranyl acetate solution, or 0.5% iron(II) chloride solution were used for contrasting. Samples were dried for 12 h at ambient temperature under sterile conditions (according to Ackermann [135], Amarillas et al. [78], Mahmoud et al. [70] procedures with modification), and visualized under JEOL JEM-1220 transmission microscope at 100,000–200,000× magnification, at a voltage of 80 kV [69]. The image background was removed in PhotoScape 3.7 software (MOII TECH, Seoul, Korea). The phages were classified according to the criteria set in the Report on the taxonomy of viruses, issued by the International Committee on Taxonomy of Viruses (ICTV) [72].

### 3.5. Extraction of Bacteriophage Genomic DNA

Bacteriophage genomic DNA was isolated from concentrated phage high-titer stock (about 10^9^ to 10^10^) using QIAamp DNA Mini Kit (Qiagen, Hilden, Germany) according to the manufacturer’s protocol. Briefly, before extracting phage DNA, phage lysates were treated with DNase and RNase A at 37 °C for 30 min to remove non-phage DNA or RNA contaminants. Then, 20 mg/mL of proteinase K and 0.5% of β-mercaptoethanol (β-ME) were added, and the phage lysate was incubated for 1 h at 55 °C. Finally, standard phenol-chloroform DNA purification with ethanol precipitation was carried out to obtain purified phage genomic DNA. DNA purity was measured by the Nanodrop ND-1000 Spectrophotometer (Thermo Fisher Scientific, Watertown, MA, USA), and DNA concentration was quantified by a Qubit 4.0 Fluorometer using the Qubit dsDNA BR Assay Kit (Invitrogen, Carlsbad, CA, USA). DNA samples were stored at −20 °C until further processing for bacteriophage whole-genome sequencing (WGS) analysis.

### 3.6. Genome Sequencing and Bioinformatics Analysis

The DNA library was prepared using the Illumina DNA Prep kit (Illumina, San Diego, CA, USA) according to the manufacturer’s protocol (number # 1000000025416v09). The magnetic bead normalization step was replaced with a manual normalization step, based on library concentration and average size as measured by the Qubit 4.0 Fluorometer with Qubit dsDNA HS Assay Kit (Thermo Fisher Scientific, Waltham, MA, USA) and the TapeStation 4200 Analyzer using the High Sensitivity D1000 ScreenTape Assay Kit (Agilent, Santa Clara, CA, USA), respectively. Whole-phage DNA was sequenced with a MiSeq next-generation sequencing platform, using the MiSeq reagent v3 (600-cycle) kit (Illumina, San Diego, CA, USA). A total of 100,000 paired reads of 300 bp without any quality trimming were assembled de novo using the CLC Genomics Workbench 7.0.3 software (CLC Bio, Denmark) with default parameters, and one contig was obtained for each phage genome. Open reading frame (ORF) prediction and annotation were done independently with the Prokka 1.12-beta software [136] and the multiPhATE software [137]. Whole-phage genome comparisons were performed using BLASTn (https://blast.ncbi.nlm.nih.gov/Blast.cgi; accessed on 1 November 2021). Circular representations of phage genomes were created using GCview [85]. Protein sequence alignments and comparisons were generated using Clustal Omega (https://www.ebi.ac.uk/Tools/msa/clustalo/; accessed 22 October 2021) and BLASTp (https://blast.ncbi.nlm.nih.gov/Blast.cg; accessed on 1 November 2021). Phage genomes were screened for virulence genes using VirulenceFinder 2.0 (https://cge.cbs.dtu.dk/services/VirulenceFinder/, last accessed on 11 July 2021), and for antibiotic resistance genes using the Comprehensive Antibiotic Resistance Database [138]. Proteomic trees of the *Enterobacter* phage KKP 3263 large terminase subunit sequence and whole-genome sequences of *Enterobacter* phage KKP 3262, *Citrobacter* phage KKP 3664, and *Serratia* phage KKP 3264 were generated using ViPTree server [84]. Comparison of ORFs from relative phages, *Enterobacter* phage KKP 3262, *Enterobacter* phage KKP3263, *Citrobacter* phage KKP 3664, and *Serratia* phage KKP 3264, was performed using the ViPTree server [84]. Assembled and annotated phage genome sequences were submitted to NCBI under the accession numbers OK210074, OK210075, OK210076, and OK210077.

### 3.7. Influence of Selected Factors on the Preservation of the Lytic Activity of Phages

This stage of the experiment aimed to determine the lytic activity of bacteriophages exposed to a wide range of temperatures and pH values. A total of 100 μL of the phage lysate was added to test tubes containing 9.9 mL of sterile physiological saline (0.85% NaCl) with a fixed active acidity (pH in the range from 3 to 12). The mixture was held at 20 °C for 1 h. To determine the activity of phage lysates at various temperatures (−20 °C, 4 °C, 20 °C, 30 °C, 40 °C, 50 °C, 60 °C, and 70 °C), 100 μL of the suspension was added to 9.9 mL of physiological saline with pH 7.0. The mixture was held for 1 h at the specified temperatures [129]. After the stage of phages’ exposure to the chemical or physical factors, a microdilution method was employed to prepare lysates with varying titers. To this end, 300 μL of the liquid LB medium and 100 μL of each 0.2 M solution of CaCl_2_ and MgSO_4_ were pipetted to sterile test tubes. Then, the mixture was completed with 100 μL of appropriately diluted lysates and 100 μL of overnight bacterial host strain (with the optical density of OD_600_~0.1). After mixing, the samples were left at 20 °C for 20 min to allow the phages’ adsorption to host cells. The lytic activity of the phages was determined with the double-layered plate method. Once the phages had adsorbed to bacterial host cells, 4 mL of soft LCA agar cooled to a temperature of 50 °C was added to the mixture (agar composition: 10 g L^−1^ of peptone, 10 g L^−1^ of sodium chloride, 5 g L^−1^ of yeast extract, and 7 g L^−1^ of agar-agar), the mixture was stirred and poured onto Petri dishes with a nutrient agar layer. Once the medium had solidified, the samples were incubated at 37 °C for 8 h (according to Shahin and Bouzari procedure [117] with modification). Afterward, plaques formed on the bacterial lawn and indicative of host cells lysis were counted, and lytic activity was computed using the formula below, considering the dilution factor, and expressed as phage titer (in plaque-forming units mL^−1^, PFU mL^−1^):M_F_ = C/R × 10, (2)
where C—number of plaques on Petri dish, (PFU); R—phage lysate dilution factor, (−); 10—factor of result conversion to 1 milliliter, (mL).

Results achieved with the double-layered plate method were presented as the percentage of activity preservation compared to the control parameters: room temperature (20 °C) and neutral pH (7.0).

### 3.8. Statistical Analysis

All of the experiments were repeated at least three times. The viable counts are expressed as the mean and standard deviation. One-way analysis of variance (ANOVA) was used to analyze the bacterial counts after bacteriophage treatment using SAS software version 9.2 (SAS Institute Inc., Cary, NC, USA). Differences in the MOI and the phage treatments were compared using Duncan’s multiple range test, and *p* < 0.05 was considered statistically significant.

## 4. Conclusions

Our study indicated that municipal sewage was a rich source of lytic phages against the dominant saprophytic bacterial microflora in minimally processed plant-based food products. Furthermore, we showed that the active acidity of the medium, compared to temperature, had a stronger influence on the ability of tested phages to infect bacterial host cells. It is noteworthy, however, that susceptibility to the effects of environmental conditions is an individual feature dependent on the type of bacteriophage. Comprehensive genomic analysis showed close genetic relatedness of the studied phages to well-known groups of lytic phages and the absence of the genes associated with lysogeny. These results indicated the virulent nature of the characterized phages in this study. Therefore, *Enterobacter* phage KKP 3263, *Citrobacter* phage KKP 3664, *Enterobacter* phage KKP 3262, and *Serratia* phage KKP 3264 have the potential to be promising novel biocontrol agents in the preservation and shelf-life extension of minimally processed plant-based food products.

## Figures and Tables

**Figure 1 ijms-22-12460-f001:**
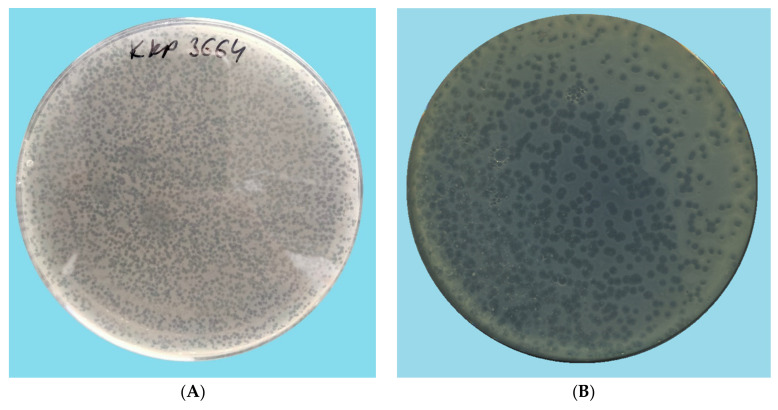
Morphology of phage plaques: (**A**) *Citrobacter* phage KKP 3664, (**B**) *Enterobacter* phage KKP 3262, (**C**) *Enterobacter* phage KKP 3263, (**D**) *Serratia* phage KKP 3264. Phages were plated in half-strength Luria-Bertani (½ LB) agar overlays with 24 h liquid culture of host bacteria. Plates were incubated at 30 °C and observed at 24 h.

**Figure 2 ijms-22-12460-f002:**
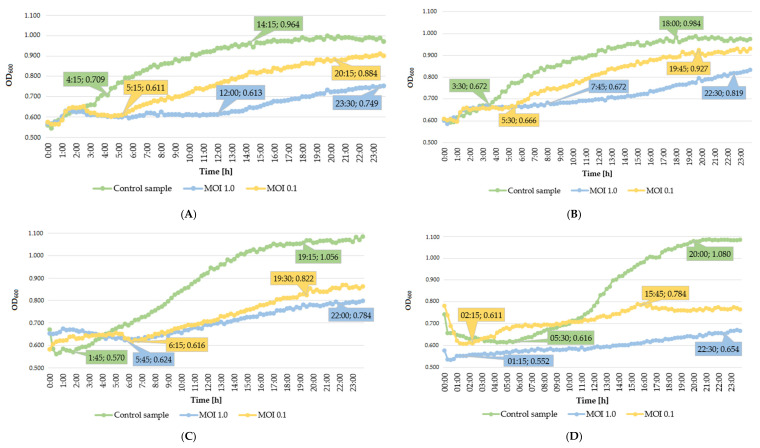
Growth curves of the bacterial strains treated with phages at infection coefficients of MOI 1.0 and MOI 0.1 compared to the control culture: (**A**) *Citrobacter freundii* KKP 3655, (**B**) *Enterobacter cloacae* KKP 3082, (**C**) *Enterobacter ludwigii* KKP 3083, (**D**) *Serratia fonticola* KKP 3084 (the figure shows also the onset and duration of the logarithmic stage of strain growth).

**Figure 3 ijms-22-12460-f003:**
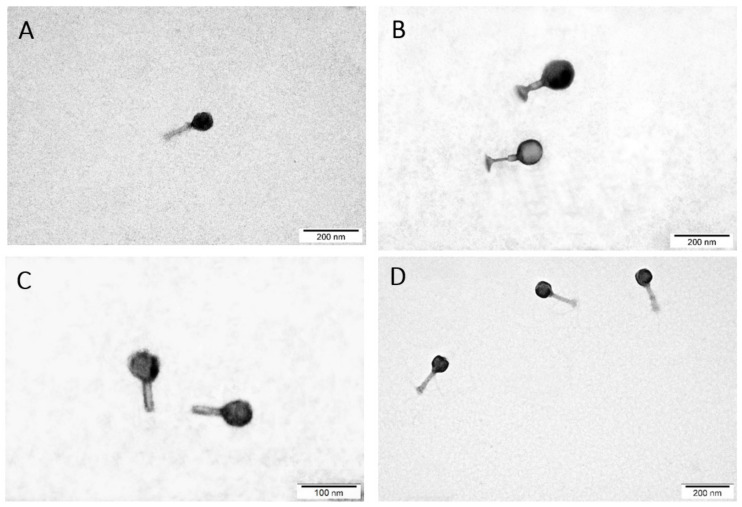
Morphology of the bacteriophages in transmission electron microscopy (TEM): (**A**) *Citrobacter* phage KKP 3664, (**B**) *Enterobacter* phage KKP 3262, (**C**) *Enterobacter* phage KKP 3263, (**D**) *Serratia* phage KKP 3264. Images of preparations were stained with the following solutions: (**A**,**C**)—2% phosphotungstic acid, (**B**)—2% uranyl acetate, and (**D**)—0.5% iron (II) chloride.

**Figure 4 ijms-22-12460-f004:**
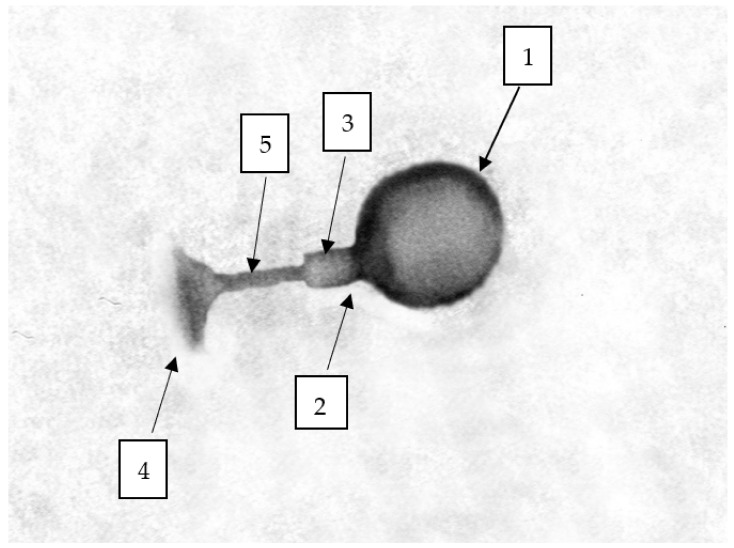
Morphology of the *Enterobacter* phage KKP 3262 (1—head, 2—neck, 3—contractile sheath, 4—baseplate, 5—tail tube).

**Figure 5 ijms-22-12460-f005:**
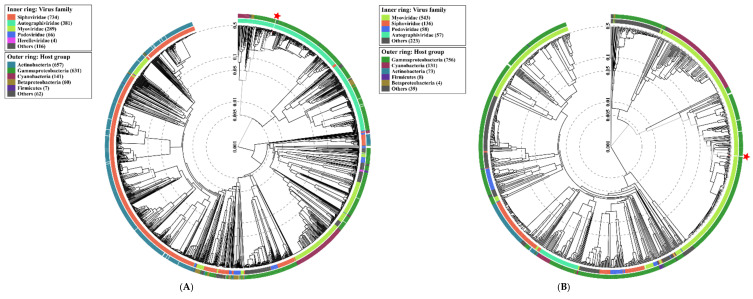
Viral proteomic trees of (**A**) *Enterobacter* phage KKP 3263, (**B**) *Enterobacter* phage KKP 3262, (**C**) *Citrobacter* phage KKP 3664, (**D**) *Serratia* phage KKP 3264, and other *Autographiviridae* and *Myoviridae* phage genomes are represented in the circular view. The branch represented studied phages is marked by an asterisk. Color rings indicate virus families (inner rings) and host groups (at a level of phylum; outer rings). These trees were calculated by BIONJ based on genomic distance matrixes, and mid-point rooted. Branch lengths are log-scaled. The sequence and taxonomic data were based on Virus-Host DB [83]. The shown trees were generated using the ViPTree server [84].

**Figure 6 ijms-22-12460-f006:**
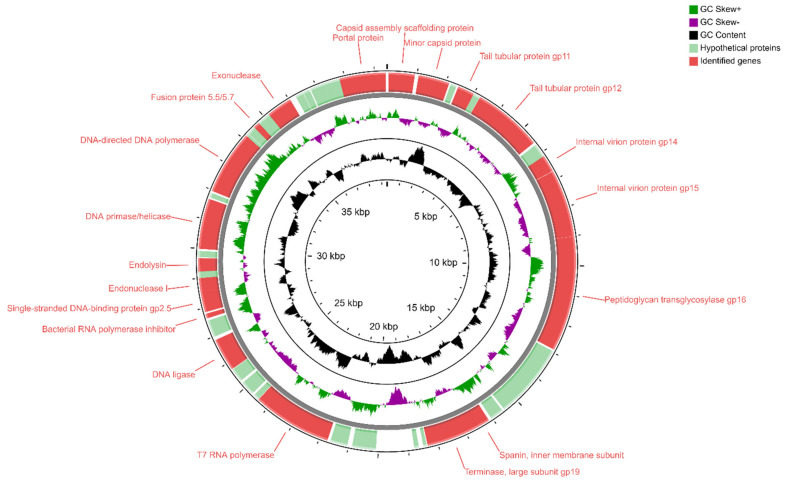
Map of the genome organization of *Enterobacter* phage KKP 3263 generated using the CGView program [85].

**Figure 7 ijms-22-12460-f007:**
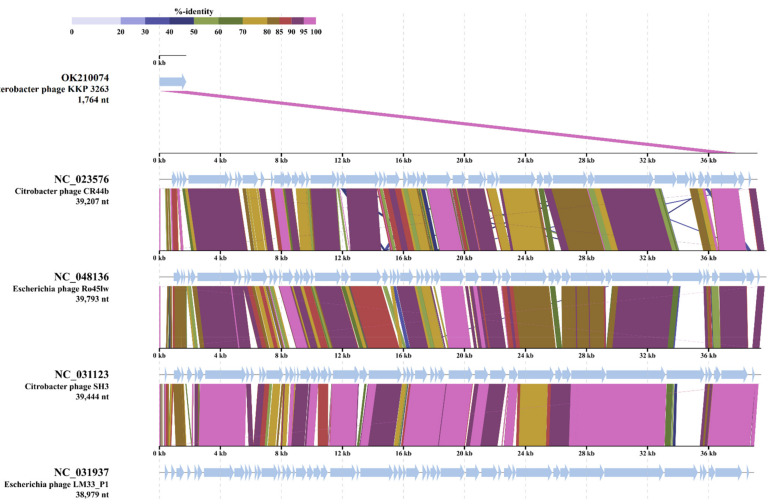
The terminase large subunit (TerL) sequence alignment of the *Enterobacter* phage KKP 3263 with four other *Autographiviridae*-related phage genomes generated by TBLASTX using ViPTree server [84]. The homologous region of the TerL detected by a TBLASTX search is connected by a segment colored based on amino acid identity. The color bar shows the % identity of TBLASTX.

**Figure 8 ijms-22-12460-f008:**
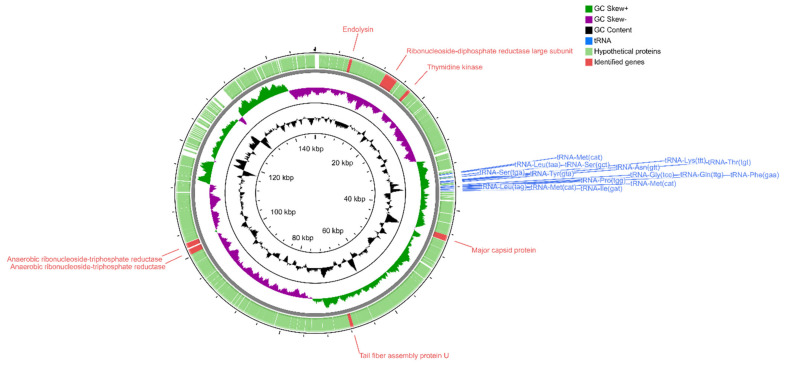
Map of the genome organization of *Serratia* phage KKP 3264 generated by using the CGView program [85].

**Figure 9 ijms-22-12460-f009:**
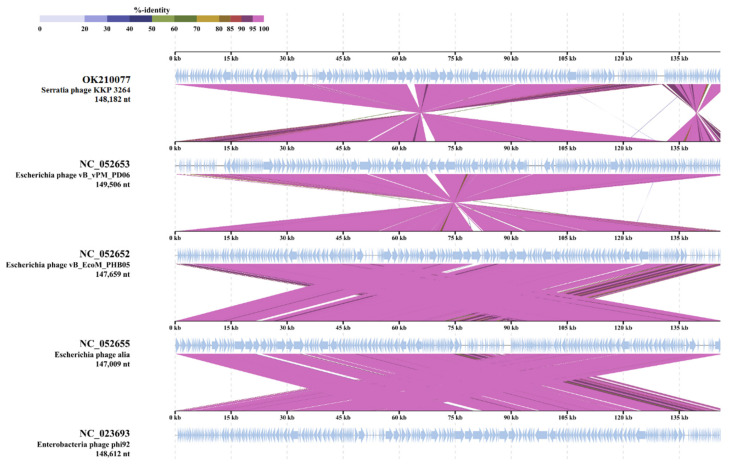
Genome sequence comparison of the *Serratia* phage KKP 3264 with four other *Myoviridae*-related phage genomes exhibiting co-linearity detected by TBLASTX using ViPTree server [84]. Homologous regions detected by a TBLASTX search are connected by segments colored based on amino acid identity. The color bar shows the % identity of TBLASTX.

**Figure 10 ijms-22-12460-f010:**
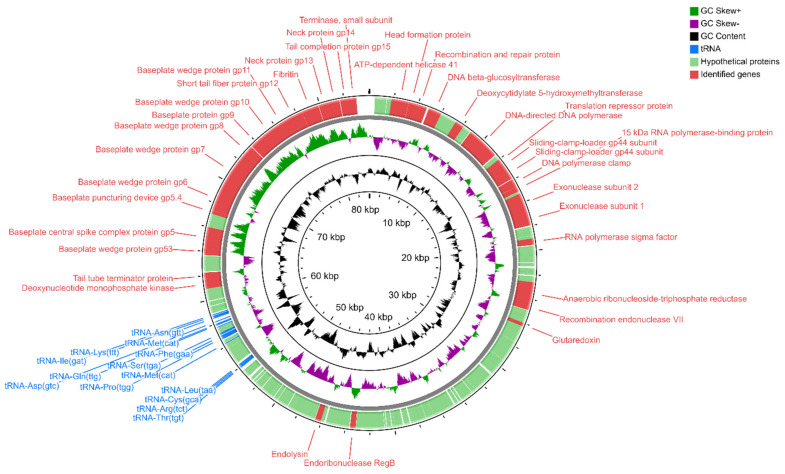
Map of the genome organization of *Enterobacter* phage KKP 3262 generated by using the CGView program [85].

**Figure 11 ijms-22-12460-f011:**
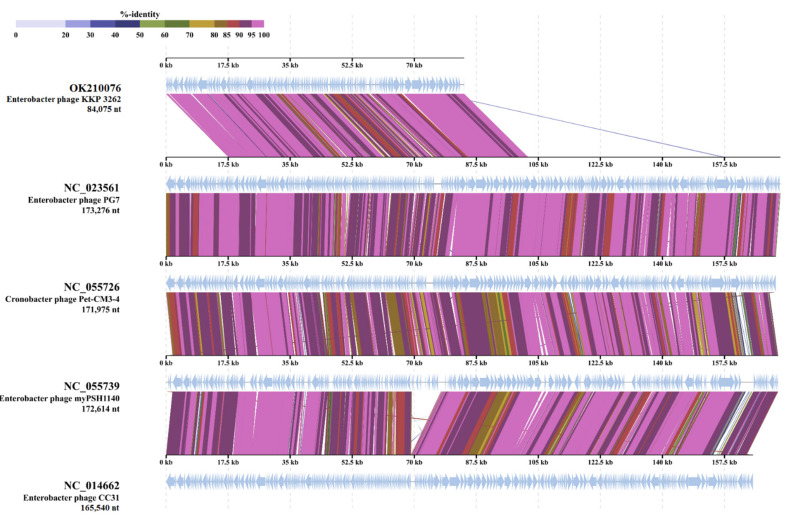
Genome sequence comparison of the *Enterobacter* phage KKP 3262 with four other *Myoviridae*-related phage genomes exhibiting co-linearity detected by TBLASTX using ViPTree server [84]. Homologous regions detected by a TBLASTX search are connected by segments colored based on amino acid identity. The color bar shows the % identity of TBLASTX.

**Figure 12 ijms-22-12460-f012:**
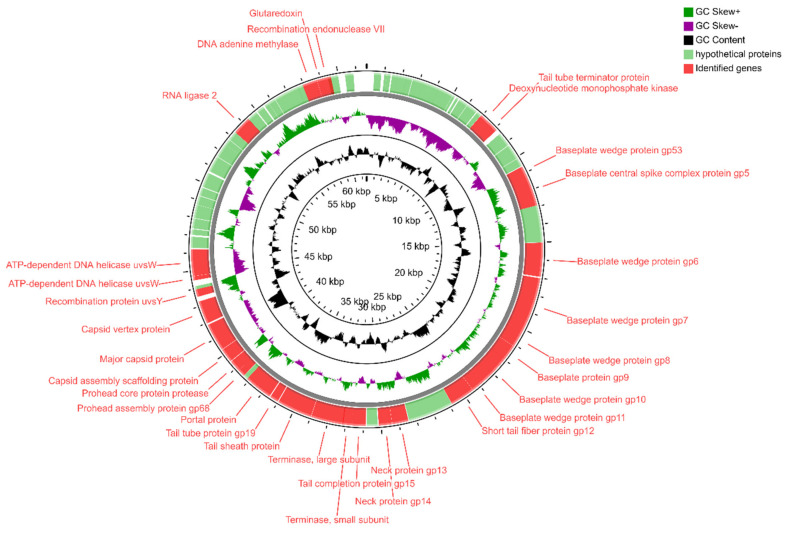
Map of the genome organization of *Citrobacter* phage KKP 3664 created by using the CGView program [85].

**Figure 13 ijms-22-12460-f013:**
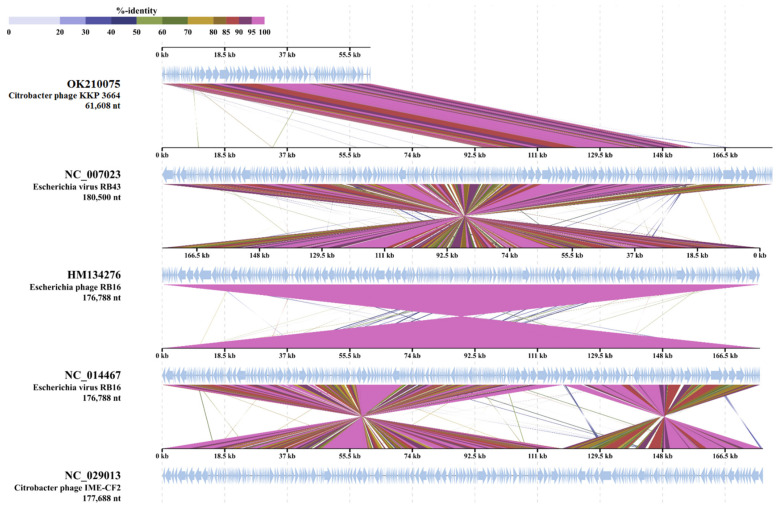
Genome sequence comparison of the *Citrobacter* phage KKP 3664 with four other *Myoviridae*-related phage genomes exhibiting co-linearity detected by TBLASTX using ViPTree server [84]. Homologous regions detected by a TBLASTX search are connected by segments colored based on amino acid identity. The color bar shows the % identity of TBLASTX.

**Figure 14 ijms-22-12460-f014:**
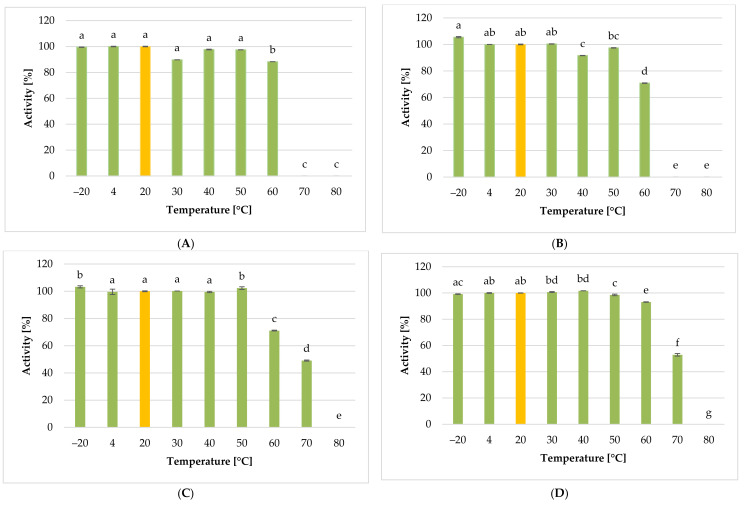
The activity of phages against tested bacterial strains at selected temperature values: (**A**) *Citrobacter* phage KKP 3664, (**B**) *Enterobacter* phage KKP 3262, (**C**) *Enterobacter* phage KKP 3263, (**D**) *Serratia* phage KKP 3264. Letters a, b, c, d, e, f, g in superscript indicate homogenous groups at a significance level of *p* < 0.05, *n* = 3.

**Figure 15 ijms-22-12460-f015:**
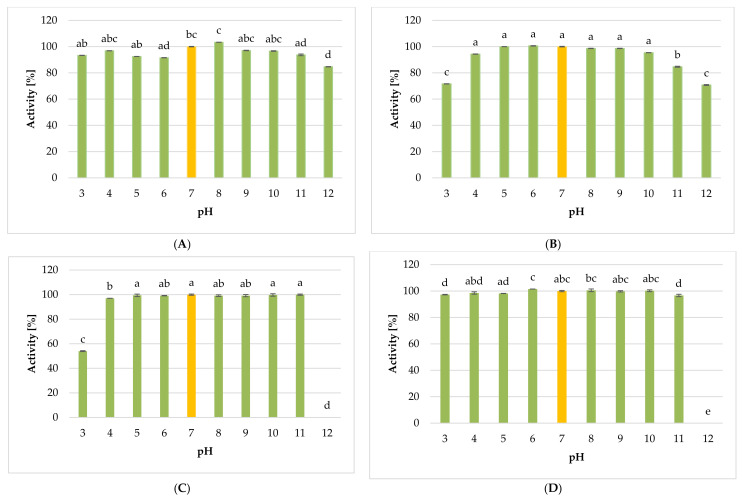
The activity of phages against tested bacterial strains in selected pH values: (**A**) *Citrobacter* phage KKP 3664, (**B**) *Enterobacter* phage KKP 3262, (**C**) *Enterobacter* phage KKP 3263, (**D**) *Serratia* phage KKP 3264. Letters a, b, c, d, e in superscript indicate homogenous groups at a significance level of *p* < 0.05, *n* = 3.

**Table 1 ijms-22-12460-t001:** Identification of selected bacterial strains.

Bacterial Strain Code	Bacterial Strain Number	Classification acc. to Bergey’s				Motility at a Temp. of 37 °C	Lactose Fermentation Capability	Bacteria Identification acc. to 16S rRNA Sequencing (GenBank No.)	Bacteria Identification acc. to MALDI-TOF MS
		Gram Staining Result	Cell Shape	Oxygen Tolerance	Spore Formation				
RUC-09	KKP 3655	Gram (−)	bacilli	relative anaerobes	non-spore-forming	motile	lactose-positive	*Citrobacter* *freundii* (MZ827001)	*Citrobacter* *freundii*
MLSC-11	KKP 3082	Gram (−)	bacilli	relative anaerobes	non-spore-forming	motile	lactose-positive	*Enterobacter* *cloacae* (MZ827006)	*Enterobacter* sp.
MLSC-21	KKP 3083	Gram (−)	bacilli	relative anaerobes	non-spore-forming	motile	lactose-positive	*Enterobacter* *ludwigii* (MZ827002)	*Klebsiella oxytoca*
MLSB-04	KKP 3084	Gram (−)	bacilli	relative anaerobes	non-spore-forming	motile	lactose-positive	*Serratia fonticola* (MZ827668)	*Serratia fonticola*

**Table 2 ijms-22-12460-t002:** The concentration of phages in lysates and optimal optical densities of bacterial cultures.

Bacterial Strain	Phage Strain	Phage Titer (M_F_) in the Lysate (PFU mL^−1^)	OD_600_ of Bacterial Culture at Which MOI 1.0
*Citrobacter freundii* KKP 3655	*Citrobacter* phage KKP 3664	6.2 × 10^9^	0.302
*Enterobacter cloacae* KKP 3082	*Enterobacter* phage KKP 3262	1.4 × 10^10^	0.262
*Enterobacter ludwigii* KKP 3083	*Enterobacter* phage KKP 3263	7.2 × 10^8^	0.234
*Serratia fonticola* KKP 3084	*Serratia* phage KKP 3264	4.4 × 10^8^	0.208

**Table 3 ijms-22-12460-t003:** Changes in the optical density of cultures of the examined strains after the addition of specific phages and values of the specific growth rate coefficient (μ).

Bacterial Strain	Control Culture		MOI 1.0		MOI 0.1	
	ΔOD	μ [h^−1^]	ΔOD	μ [h^−1^]	ΔOD	μ [h^−1^]
*Citrobacter freundii* KKP 3655	0.255	0.031	0.136	0.017	0.273	0.025
*Enterobacter cloacae* KKP 3082	0.312	0.026	0.147	0.019	0.261	0.023
*Enterobacter ludwigii* KKP 3083	0.486	0.035	0.160	0.014	0.206	0.022
*Serratia fonticola* KKP 3084	0.464	0.039	0.102	0.008	0.173	0.019

## Data Availability

Not applicable.

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
