# Peer review of "Characterization and Genome Study of Novel Lytic Bacteriophages against Prevailing Saprophytic Bacterial Microflora of Minimally Processed Plant-Based Food Products"

_ijms, 2021, doi:10.3390/ijms222212460_

Round 1

Reviewer 1 Report

The discovery and investigation of 4 novel phage is a good contribution to the food safety field.  The authors have conducted a thorough characterization of all phages and while the genomic characterization is very complete and interesting, could have included more microbiological assays to add to the case for these phages being used in food biocontrol.  

Section 2.1 - Please discuss whether the bacteria identified in this study are typically a problem in food safety applications. Do they cause illness, etc?

Line 34 - No antibiotic resistance genes, virulence factors, integrase, recombinase, or repressors, that are the main markers of lysogenic viruses. - Not a complete sentence. Please check throughout the paper for sentence fragments and proper English. 

Line 93 - Remove ) after P100

Line 97- Micros should be Micreos

Line 100 - significant reduction should be significantly reduce

Line 163 -173 - This is background information that should be in the introduction instead of the results.

Figure 2 - Please indicate what wavelength the OD was determined on the y axis. Please also include error bars. 

Line 342- contacted sheath should be contractile sheath

Table 4 is not really necessary and does not add anything to this manuscript. The details are all in the above text. 

Section 2.5 Please remove this section. It is not necessary since you are discussing the results in the next two sections.  If you would like to keep it please move to the introduction as it is background information and does not contain any results. 

Line 902 - To prevent the activation of resistance mechanisms to phage infection by the hosts, desired will be these phages that will be capable of lytic development. This sentence is not proper English.

Author Response

Manuscript 1470867

Response to Reviewers

Dear Editor and Reviewers,

Thank you for allowing us to submit a revised draft of the manuscript “Characterization and genome study of novel lytic bacteriophages against prevailing saprophytic bacterial microflora of minimally processed plant–food products” to publication in the IJMS. We appreciate the time and effort that you and the reviewers dedicated to providing feedback on our manuscript and we are grateful for the insightful comments and valuable improvements to our paper. We have incorporated most of the suggestions made by the reviewers. Those changes are done using the “Track Changes” function within the manuscript. Please see below, in blue, for a point-by-point response to the reviewers’ comments and concerns. All page numbers refer to the revised manuscript file with tracked changes.

Reviewer 2 Report

Comments to the paper “Characterization and genome study of a novel lytic bacteriophage against prevailing saprophytic bacterial microflora of minimally processed plant–food products” by Wójcicki et al.

General comments: it is a while since the use of bacteriophages as potential biological control agents in microbiological food safety has been proposed. Actually, bacteriophages are being used as an alternative to the use of antibiotics since Russia was part of U.R.S.S.. However, just a few study were conducted on this topic and even less on vegetables intended for human consumption. Thus, the topic of high impact and, due to the high resistance to antibiotics registered around the world, indeed it represents a hot topic. The speculations seems to be overestimated, the work is not applicative on bacteria being inhibited in vivo (e.g. directly in salads).

The methodology is sound and the results clearly presented.

The topic falls within International Journal of Molecular Sciences scopes and the manuscript is of interest for IJMS readers. I am not the best person to judge the English style, but the manuscript is not easy to follow and it does not seem to be clearly written. A deep reading by a native speaker is advisable.

Specific points:

  • “respectively” related to what? “…the size was 39,418 bp for KKP 3263…” and so on.
  • L34-35. “No antibiotic resistance genes, virulence factors, integrase, recombinase, or repressors, that are the main markers of lysogenic viruses.” This sentence is incomplete.
  • L38-42. Five pages of speculation it too much for a study partial evaluating the efficacy of the phages.
  • L48-77. This section would make sense for an applicative work on vegetable safety. The authors did not test the efficacy of what they suppose in vivo. This introduce the readers to a work where phages are applied to control pathogens in salads, but it simply evaluates the inhibition of pure cultures of bacteria
  • The abstract is not descriptive enough. “The bacteriophages retained their lytic activity in a wide range of temperatures (from –20 °C to 50 °C) and active acidity values (pH from 4 to 11). The active acidity of the environment had the strongest impact on phages' capability for infecting bacterial host cells.” Please provide numerical data (in form of %, concentrations, activity, etc.). As is, this section, that should stand alone independently of the text, lacks of the more relevant data.
  • Keywords: minimally processed food and bio-preservation cannot be used as keywords, since the authors did not proven the in vivo effectiveness of the system proposed.

Author Response

(The authors gave the same response as above.)
